*Method*

# CIGAR-seq, a CRISPR/Cas-based method for unbiased screening of novel mRNA modification regulators

Liang Fang[1,2,*,†] [ID], Wen Wang[1,3,†], Guipeng Li[1,2,†], Li Zhang[1], Jun Li[1], Diwen Gan[1], Jiao Yang[1], Yisen Tang[1], Zewen Ding[1], Min Zhang[1], Wenhao Zhang[1], Daqi Deng[1], Zhengyu Song[1], Qionghua Zhu[1], Huanhuan Cui[1,2], Yuhui Hu[1] [ID] & Wei Chen[1,2,**] [ID]

## Abstract

Cellular RNA is decorated with over 170 types of chemical modifications. Many modifications in mRNA, including $m^6A$ and $m^5C$, have been associated with critical cellular functions under physiological and/or pathological conditions. To understand the biological functions of these modifications, it is vital to identify the regulators that modulate the modification rate. However, a high-throughput method for unbiased screening of these regulators is so far lacking. Here, we report such a method combining pooled CRISPR screen and reporters with RNA modification readout, termed CRISPR integrated gRNA and reporter sequencing (CIGAR-seq). Using CIGAR-seq, we discovered NSUN6 as a novel mRNA $m^5C$ methyltransferase. Subsequent mRNA bisulfite sequencing in HAP1 cells without or with NSUN6 and/or NSUN2 knockout showed that NSUN6 and NSUN2 worked on non-overlapping subsets of mRNA $m^5C$ sites and together contributed to almost all the $m^5C$ modification in mRNA. Finally, using $m^1A$ as an example, we demonstrated that CIGAR-seq can be easily adapted for identifying regulators of other mRNA modification.

**Keywords** CIGAR-seq; $m^5C$ modification; mRNA modification; NSUN6; pooled CRISPR screen

**Subject Categories** Methods & Resources; RNA Biology

**Mol Syst Biol. (2020) 16: e10025**

## Introduction

Cellular RNAs can be chemically modified in over a hundred different ways, and such modifications have been associated with diverse cellular functions under physiological and/or pathological conditions (Machnicka *et al*, 2013; Roundtree *et al*, 2017). To epitranscriptomic regulation, each modification needs its distinct deposition, removal, and recognition factors (termed "writers", "erasers", and "readers", respectively). Yet, comparing to the ever-expanding techniques on detecting RNA modifications (Zhao *et al*, 2020), the methods to systematically identify writers and erasers of RNA modifications are rather limited. For instance, the first N6-adenosine ($m^6A$) methyltransferase METTL3 was identified through a combination of in vitro assay, conventional chromatography, electrophoresis, and microsequencing (Bokar *et al*, 1994; Bokar *et al*, 1997); and METTL14, a key component of $m^6A$ methyltransferase complex, was discovered through the phylogenetic analysis based on METTL3 (Wang *et al*, 2014). In general, the first strategy is less efficient and may have assay-specific bias, while the second strategy relies on the prior knowledge of related molecule(s). So far, unbiased method to screen for novel regulators of RNA modifications is still lacking.

Recently, the rapid development of CRISPR-based gene manipulation provides a new paradigm for high-throughput and genome-wide functional screening. Pooled CRISPR screen outperforms array-based screen by its scalability and low cost, however, was largely restricted to standard readouts, including survival, proliferation, and FACS-sortable markers (Hanna & Doench, 2020). Most recently, combining with microscopy-based approaches, CRISPR screen enabled the association of subcellular phenotypes with perturbation of specific gene(s) (Wheeler *et al*, 2020; preprint: Yan *et al*, 2020). In studying regulation of gene expression, Perturb-seq, CRISP-seq, and CROP-seq, which combine CRISPR-based gene editing with single-cell mRNA sequencing, allowed transcriptome profile to serve as comprehensive molecular readout (Adamson *et al*, 2016; Dixit *et al*, 2016; Datlinger *et al*, 2017), but often with limited throughput. Until now, pooled CRISPR screen with epitranscriptomic readout has not yet been developed.

One important RNA modification, 5-methylcytosine ($m^5C$), was first identified in stable and highly abundant tRNA and rRNA (Helm,

1 Department of Biology, Southern University of Science and Technology, Shenzhen, Guangdong, China
2 Academy for Advanced Interdisciplinary Studies, Southern University of Science and Technology, Shenzhen, Guangdong, China
3 Harbin Institute of Technology, Harbin, Heilongjiang, China
*Corresponding author. Tel: +86 138 23288350; E-mail: fangl@sustech.edu.cn
**Corresponding author. Tel: +86 755 88018449; E-mail: chenw@sustech.edu.cn
†These authors contributed equally to this work

2006; Agris, 2008; Schaefer *et al*, 2009). Subsequently, many novel m[5]C sites in mRNA were discovered by using next-generation sequencing-based methods, including mRNA bisulfite sequencing (mRNA-BisSeq) (Schaefer *et al*, 2009; Squires *et al*, 2012), m[5]C-RNA immunoprecipitation (RIP) (Edelheit *et al*, 2013), 5-azacytidine-mediated RNA immunoprecipitation (Aza-IP) (Khoddami & Cairns, 2013), and methylation-individual-nucleotide-resolution crosslinking and immunoprecipitation (miCLIP) (Hussain *et al*, 2013). The m[5]C modification has been reported to regulate the structure, stability, and translation of mRNAs (Luo *et al*, 2016; Li *et al*, 2017a; Guallar *et al*, 2018; Shen *et al*, 2018; Schumann *et al*, 2020) and be catalyzed by NOP2/Sun RNA methyltransferase family member 2 (NSUN2) (Khoddami & Cairns, 2013; David *et al*, 2017; Yang *et al*, 2017). However, recent studies have shown that, even after NSUN2 knockout (KO), a significant number of m[5]C sites in mRNA remained methylated (Huang *et al*, 2019; Trixl & Lusser, 2019), suggesting the existence of additional methyltransferase(s) involved in mRNA m[5]C modification. To fully appreciate the function of this modification, it would be important to identify the remaining methyltransferase(s).

Here, we report a method combining pooled CRISPR screen and a reporter with epitranscriptomic readout, termed CRISPR integrated gRNA and reporter sequencing (CIGAR-seq). Using CIGAR-seq with a reporter containing a m[5]C modification site, we screened through a gRNA library targeting 829 RNA-binding proteins and identified NSUN6 as a novel m[5]C writer of mRNA. mRNA-BisSeq in HAP1 cells without or with NSUN6 and/or NSUN2 knockout showed NSUN6 and NSUN2 worked on non-overlapping subsets of mRNA m[5]C sites and together contributed to almost all the m[5]C modification in mRNA. Finally, using m[1]A as an example, we demonstrated that CIGAR-seq can be easily adapted for studying other mRNA modification.

# Results

## CIGAR-seq: Pooled CRISPR screening with a epitranscriptomic readout

In CIGAR-seq, to integrate pooled CRISPR screening with a epitranscriptomic readout, here more specifically m[5]C modification readout, we adopted the previously developed CROP-seq method (Datlinger *et al*, 2017) and replaced the WPRE cassette on the original vector by an endogenous m[5]C site with its flanking region (Fig 1A). Thereby, the mRNA molecules transcribed from this lentiviral vector contain a selection marker followed by an endogenous m[5]C site, a U6 promoter and a gRNA sequence. To detect the m[5]C level in the gRNA sequence-containing transcripts, total mRNA was firstly subjected to bisulfite treatment followed by reverse transcription. Subsequently, a primer pair flanking the m[5]C site and gRNA sequence was used to amplify the region for Sanger or next-generation sequencing (Fig 1A). In this way, the methylation level of the m[5]C reporter site can be measured and associated with the gRNA targeting a specific gene.

As a proof-of-concept experiment, a known NSUN2-dependent m[5]C site in FAM129B (also known as NIBAN2) gene was cloned into CIGAR-seq vector, together with a control gRNA without any target gene or a gRNA targeting NSUN2 (Fig 1B, upper panel).

Seven days after transduction into Cas9-expressing HAP1 cells, m[5]C modification level on the reporter transcripts was measured. The result demonstrated that, upon NSUN2 perturbation (Fig EV1A), the m[5]C modification rate reduced significantly in the reporter containing NSUN2 targeting gRNAs (Fig 1B, lower panel), whereas the modification remained intact in the reporter containing the control gRNA (Fig 1B, upper panel).

## CIGAR-seq identified NSUN6 as a novel mRNA methyltransferase

As suggested by previous studies that large amount of m[5]C sites in mRNA remained methylated after NSUN2 knockout (Huang *et al*, 2019; Trixl & Lusser, 2019), we sought to utilize CIGAR-seq to identify gene(s) that mediate the m[5]C modification on NSUN2-independent sites. First, to determine the NSUN2-independent m[5]C sites, we established NSUN2 knockout (NSUN2-KO) HAP1 cells (Fig EV2A), and performed mRNA bisulfite sequencing (mRNA-BisSeq) in wild-type as well as NSUN2-KO cells. Following bioinformatic pipeline proposed by Huang *et al* (2019), a set of 208 m[5]C sites was identified in wild-type HAP1 cells (Materials and Methods), only 90 (43.3%) of which showed significantly reduced m[5]C level in NSUN2-KO cells (Fig EV1B).

We then chose a NSUN2-independent site in the 3'UTR of FURIN gene with a high m[5]C modification rate as the reporter site for CIGAR-seq. Meanwhile, a gRNA library targeting 829 RNA-binding proteins (RBP) was synthesized (Dataset EV1). To establish the CIGAR-seq vector pool for the genetic screen, the gRNA library was firstly cloned into the vector followed by the insertion of the FURIN m[5]C site with its flanking genomic region (Materials and Methods). Cas9-expressing HAP1 cells were then transduced with CIGAR-seq virus pool (Fig 1C, Materials and Methods). Seven days after transduction, cells were collected and subjected to RNA extraction. Enriched polyA RNA was then bisulfite-treated, reveres-transcribed and PCR-amplified using primers flanking the m[5]C site and gRNA sequence to generate next-generation sequencing (NGS) library (Materials and Methods). After pair-end sequencing and data processing, 811 genes were detected with at least one gRNA, of which 782 genes had at least two gRNAs (Fig EV1C). The m[5]C modification rate of reporter site was calculated for each gRNA. While the median m[5]C modification rates of gRNA-associated reporter sites were around 93.5%, a small part of gRNAs showed significantly reduced m[5]C rates (Fig EV1D). To prioritize the candidate genes, Stouffer's method was used to calculate the combined P-value based on the gRNAs targeting the same gene (Materials and Methods). As shown in Fig 1D, it turned out that NSUN6, a member of NOL1/NOP2/sun domain (NSUN) family, was identified as the best hit with all six gRNAs decreasing the m[5]C level effectively (Fig 1D and Dataset EV2). Interestingly, NSUN6 was previously reported to introduce the m[5]C in tRNA (Li *et al*, 2019). However, two previous studies did not show significant m[5]C changes in mRNA after NSUN6 perturbation in HeLa cells (Yang *et al*, 2017; Huang *et al*, 2019), which might be due to the incomplete gene silencing mediated by siRNA.

To validate our result, a gRNA targeting NSUN6 was inserted into the CIGAR-seq vector containing the FURIN m[5]C site. As shown in Fig 1E, perturbation of NSUN6 (Fig EV3A) indeed reduced the m[5]C level in both reporter mRNA and endogenous NSUN2-independent m[5]C site in RPSA gene (Fig 1E, left and

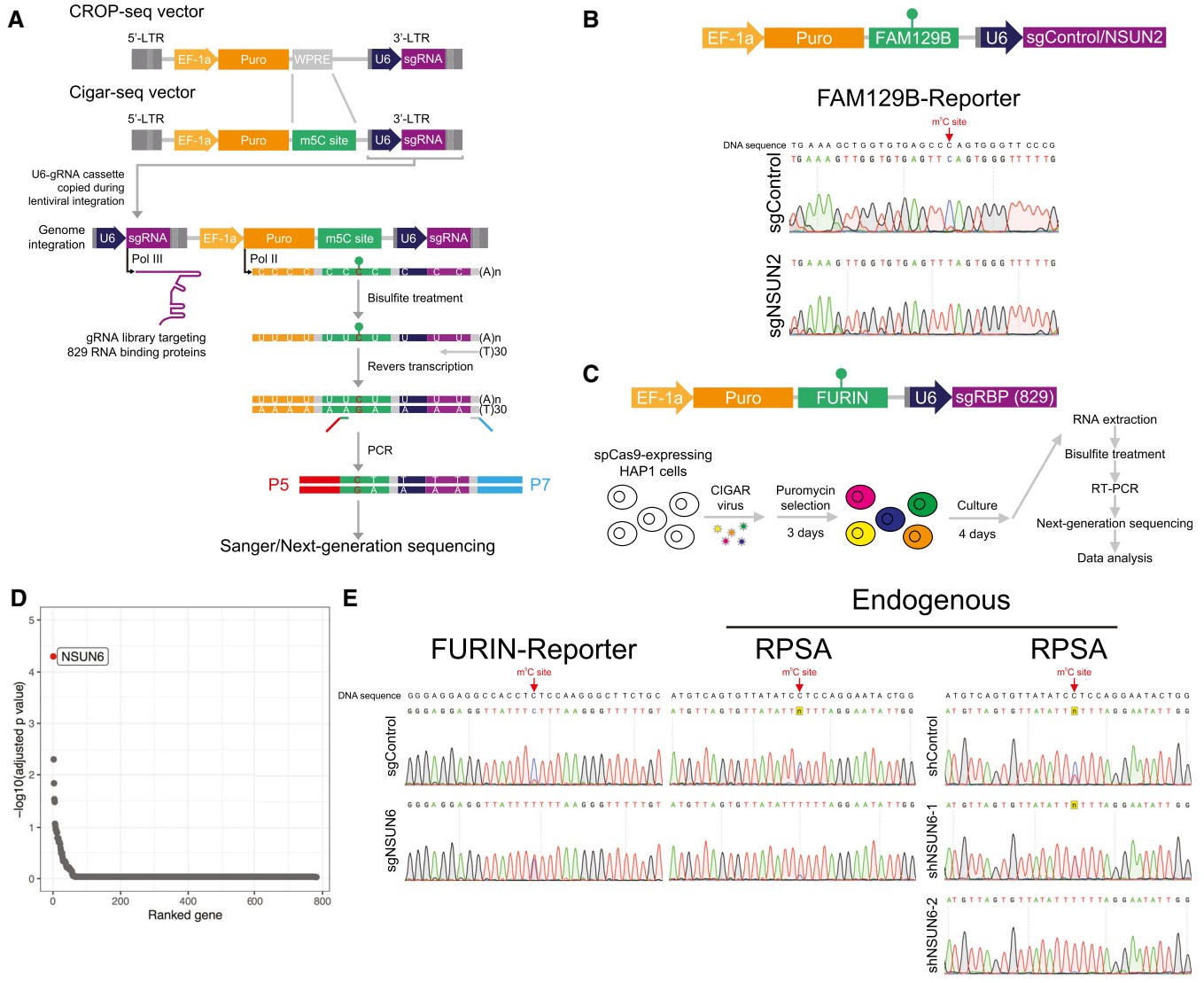

**Figure 1.  CIGAR-seq identified NSUN6 as a novel mRNA methyltransferase.**

A   An illustration of CIGAR-seq method in studying m5C modification. The WPRE cassette on the original CROP-seq vector was replaced by an endogenous m5C site with its flanking region. To measure the m5C level of the reporter site in gRNA sequence-containing transcripts, mRNA was subjected to bisulfite treatment followed by reverse transcription. Then, a primer pair flanking the m5C site and gRNA sequence was used to amplify the region for subsequent Sanger or next-generation sequencing.

B   Validation of CIGAR-seq method using a m5C reporter site derived from FAM129B gene with a control gRNA without target genes and a gRNA targeting NSUN2. Upon NSUN2 knockout, the m5C modification is diminished in the FAM129B reporter mRNA, whereas the modification remains intact in control knockout cells.

C   The application of CIGAR-seq in screening for regulators of m5C sites. Cas9-expressing HAP1 cells were transduced with viral particles that express Cigar vectors combining NSUN2-independent m5C reporter sites derived from FURIN gene and a gRNA library targeting 829 RBPs. Seven days after transduction, enriched polyA RNA was bisulfite-treated, reveres-transcribed, and PCR-amplified using primers flanking the m5C site and gRNA sequence to generate NGS library.

D   The rank of genes whose knockout reduced m5C modification rate of the reporter site. To identify the high-confident candidate genes, information of multiple gRNAs of the same genes was combined using the Stouffer's method, then a combined P-value for each gene was calculated using the weighted version of Stouffer's method. NSUN6 was the top hit.

E   Validation of NSUN6 as a mRNA m5C methyltransferase. Knockout as well as knock-down NSUN6 reduced the m5C level in both FURIN m5C reporter transcripts and endogenous NSUN2-independent m5C sites in RPSA gene.

middle panels). Furthermore, to rule out the potential off-target effect of gRNA, we repressed NSUN6 expression using shRNA (Fig EV3B). Again, the m5C level at the endogenous site was also reduced in cells with NSUN6 repression (Fig 1E, right panel). Together, these results confirmed NSUN6 as a bona fide mRNA m5C methyltransferase.

In addition to NSUN6, we selected another two candidate genes, EIF3J and ZCCHC11 with multiple gRNAs (five and six, respectively, Dataset EV2) showing inhibitory effect on m5C modification for validation. However, knockout of neither ZCCHC11 nor EIF3J could reduce the modification rate of the m5C site in the reporter transcripts.

### Global profiling of NSUN6-dependent m⁵C sites

To globally characterize NSUN6-dependent m⁵C sites, we established NSUN6 knockout (NSUN6-KO) HAP1 cells (Fig EV2B) and performed mRNA-BisSeq. Of 208 m⁵C sites identified in wild-type HAP1 cells, 65 (31.2%) showed significant reduction at m⁵C level in NSUN6-KO cells (Fig 2A). To illustrate the features of sequence flanking NSUN6-dependent m⁵C sites in HAP1 cells, motif analysis was performed based on the upstream and downstream 10 nucleotide sequences flanking the m⁵C sites. As shown in Fig 2B, NSUN6-dependent m⁵C sites were embedded in slightly GC-rich environments with a strongly enriched TCCA motif at 3′ of m⁵C sites. Previously, a similar 3′ TCCA motif was also found at NSUN6 target sites in tRNAs (Li et al, 2019) and has also been proposed as sequence motif around NSUN2-independent sites in another study (Huang et al, 2019). In comparison, the sequence feature around NSUN2-dependent m⁵C sites is distinct, which is enrich for 3′ NGGG motif (Yang et al, 2017; Huang et al, 2019).

### Contribution of NSUN6 and NSUN2 to the mRNA m⁵C modification

We then evaluated the relative contribution of NSUN6 and NSUN2 to the global mRNA m⁵C modification. First, comparing

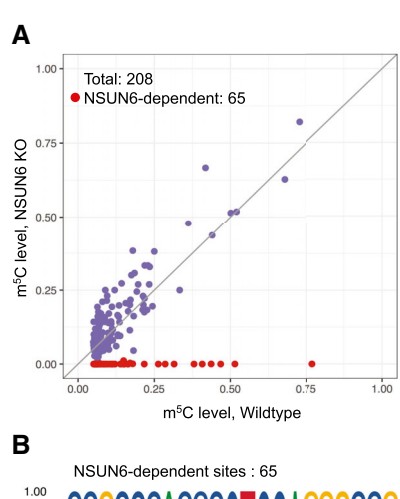

**A**

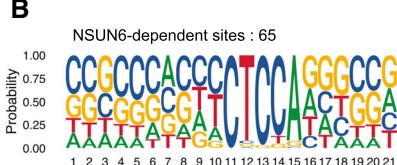

**B**

### Figure 2. Global profiling of NSUN6-dependent m⁵C sites.

A   mRNA bisulfite sequencing revealed NSUN6-dependent m⁵C sites in HAP1 cells. Of 208 m⁵C sites identified in wild-type cells, 65 showed significantly reduced modification in NSUN6 knocked out cells. X- and Y-axis represented the modification rate in wild-type and NSUN6 knocked out HAP1 cells, respectively. The gray line represents the diagonal line, along which the modification rate is equal between wild-type and NSUN6 knockout cells.

B   The sequence features of NSUN6-dependent m⁵C sites in HAP1 cells. A strong 3′ TCCA motif was found in NSUN6-dependent sites.

between NSUN2- and NSUN6-dependent m⁵C sites, as shown in Fig 3A, these sites were largely non-overlapping, suggesting their non-redundant biological functions. Then, to further examine whether NSUN2 and NSUN6 together are responsible for all mRNA m⁵C modifications, NSUN2 and NSUN6 double KO (NSUN2/6-dKO) HAP1 cells were established (Fig EV2C) and subjected to mRNA-BisSeq analysis. As shown in Fig 3B, the modification of m⁵C sites depend only on NSUN6 or NSUN2 (62 and 87, respectively) were also abolished in NSUN2/6-dKO cells. While NSUN6-dependent sites were strongly enriched for 3′ TCCA motif as shown earlier, NSUN2-dependent sites were enriched for 3′ NGGG motif as previously reported (Yang et al, 2017; Huang et al, 2019) (Fig 3C). Furthermore, we carefully examined the three sites that showed dependence on both NSUN6 and NSUN2, as well as the 56 sites that were independent of both NSUN6 and NSUN2. Comparing to the other three groups, the group of three overlapping sites had very low m⁵C level (Fig 3D). In addition, the m⁵C sites in ANGEL1 and ZNF707 possessed a 3′ TCCA and a 3′ AGGG motif, respectively (Fig EV4A and B), suggesting they are very likely a NSUN6- and a NSUN2-dependent site, respectively, but with low m⁵C level that led to false negative findings in the mRNA-BisSeq analysis of some but not all the samples. The remaining m⁵C site in STRN4 was embedded within a cluster of "pseudo" m⁵C sites (Fig EV4C), which was highly likely an artifact due to the incomplete bisulfite conversion as suggested before (Haag et al, 2015; Huang et al, 2019). Similarly, the group of 56 NSUN2/6-independent sites was also highly enriched for such clusters of pseudo m⁵C sites: 52 sites had at least one pseudo m⁵C site in vicinity (Fig EV5). The remaining four sites all had very low m⁵C level.

Of note, here, to characterize the NSUN6- and NSUN2-in/dependent m⁵C sites, we set stringent criteria to avoid potential false discovery of m⁵C sites due to incomplete bisulfite conversion (Materials and Methods). Whereas such stringent criteria would assure the high specificity of our findings, we would likely also miss some of the true m⁵C sites, particularly those with low modification rate.

To explore the modification rate of NSUN6/2-dependent m⁵C sites across different tissues, we resorted to mRNA-BisSeq data from a previous study (Huang et al, 2019). As shown in Fig 3E, m⁵C modification on 47 NSUN6- and 66 NSUN2-dependent m⁵C sites could also be observed in other human tissue(s). While the modification rate of NSUN6-dependent sites was by and large highest in liver, NSUN2-dependent ones did not show such tissue biases (Fig 3E).

### CIGAR-seq could be used for the study of other mRNA modification

Finally, to explore the potential application of CIGAR-seq in the study of other mRNA modifications, we turned to N-1-methyladenosine (m¹A). As *N*-1-methyladenosine (m¹A) can cause misincorporation during cDNA synthesis (Hauenschild et al, 2015), its modification can be detected by direct cDNA sequencing. Similar as the previous NSUN2 proof-of-concept experiment, we chose a well-characterized m¹A site from MALAT1, which is known to be modified by TRMT6/TRMT61A complex (Dominissini et al, 2016; Li et al, 2017b; Safra et al, 2017). We cloned the site and its flanking

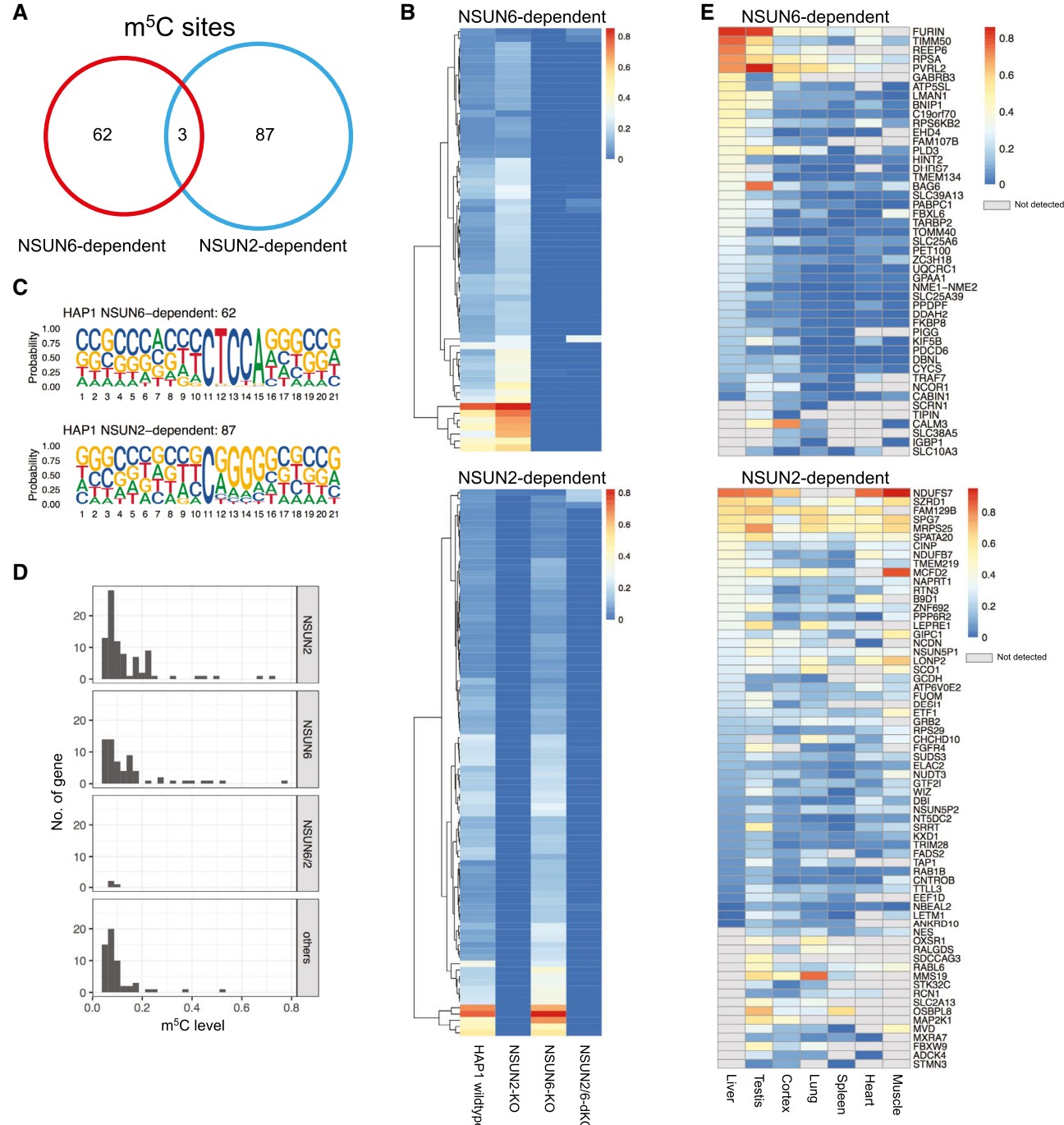

**Figure 3. Comparison between NSUN6- and NSUN2-dependent m⁵C modification sites.**

A  The largely non-overlapping pattern between NSUN2- and NSUN6-dependent m⁵C sites.
B  Heat map showing the m⁵C modification rate in wild-type, NSUN2-KO, NSUN6-KO, and NSUN2/6-dKO HAP1 cells for NSUN6- (upper panel) and NSUN2-dependent sites (lower panel), respectively.
C  The sequence features of NSUN6-only- (upper panel) and NSUN2-only-dependent m⁵C sites (lower panel) in HAP1 cells. While NSUN6-dependent sites were strongly enriched for 3′ TCCA motif, NSUN2-dependent sites were enriched for 3′ NGGG motif.
D  The modification rate of 4 groups of m⁵C sites that showed different dependence. Comparing to the other three groups, the group of three overlapping sites showed very low m⁵C level.
E  Modification rate of NSUN6- and NSUN2-dependent m⁵C sites across different tissues.

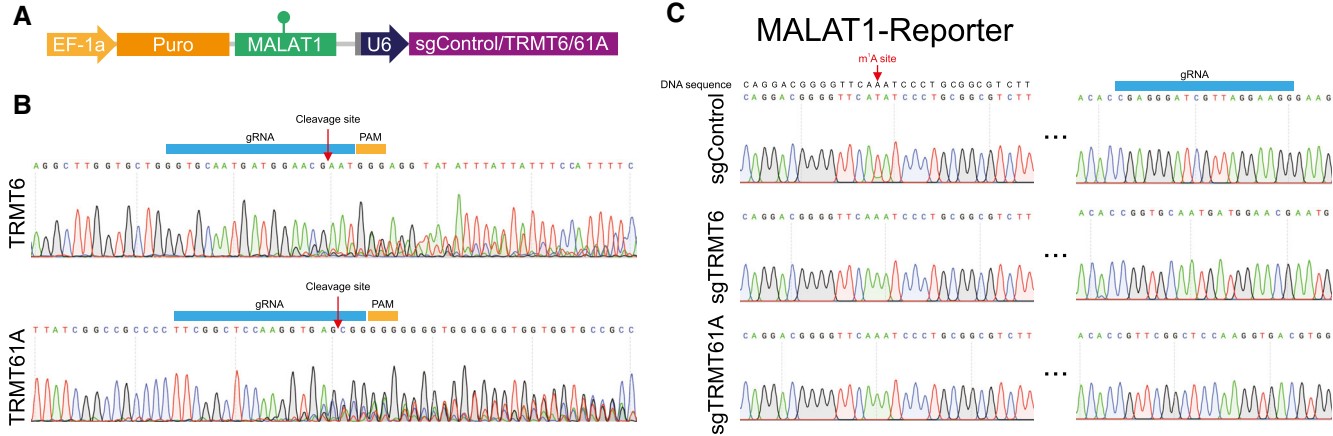

**Figure 4.  Exemplar application of CIGAR-seq in the study of m¹A modification.**

A       An illustration of CIGAR-seq vector designed for m¹A modification. A known TRMT6/61A complex-dependent m¹C site in MALAT1 gene was cloned into CIGAR-seq vector, together with control gRNA as well as gRNAs targeting TRMT6 and TRMT61A, respectively.

B, C   Upon perturbation of either TRMT6 or TRMT61A in HAP1 cells, the m¹A modification was completed abolished in m¹A reporter site, whereas the modification remains intact in control knockout cells.

region into CIGAR-seq vector with a control gRNA and two gRNAs targeting TRMT6/TRMT61A complex, respectively (Fig 4A). As shown in Fig 4C, in HAP1 cells with perturbation of either TRMT6 or TRMT61A (Fig 4B), the m¹A modification of the reporter site was completely abolished, whereas in cells transduced with control gRNAs, the modification remains intact.

## Discussion

Combining pooled CRISPR screening strategy and a reporter with epitranscriptomic readout, CIGAR-seq for the first time enables the unbiased screening for novel regulators of mRNA modifications. In this study, we demonstrated its power in identification of NSUN6 as a novel mRNA m⁵C methyltransferase. In addition, we also showed its potential application in studying m¹A modification. Integrating additional modification readout strategies into our pipeline, it could be further adapted to investigate other modifications. For instance, we can use CIGAR-seq to search for potential regulators of RNA editing by simply reading the A-G or C-T changes in the cDNA sequence reads derived from A-I or C-U RNA editing reporters. m⁶A-iCLIP (Linder *et al*, 2015) or SELECT (Xiao *et al*, 2018) method, which were used to measure the modification rate of individual m⁶A site, could also be integrated into our CIGAR-seq in analyzing m⁶A regulators. Furthermore, changing reporters to those with other regulatory readout, for example alternative splicing or alternative polyadenylation pattern, the potential application of CIGAR-seq could be easily extended to screen for factors involved in diverse post-transcriptional regulations. There, given the readout is based on directly measuring the reporter-derived RNAs, CIGAR-seq would be in principle superior to current fluorescence-reporter-FACS-based screening strategies.

Like any other high-throughput assays, CIGAR-seq has also its own sensitivity and specificity issues, which could be affected by

the choice of reporter and CRISPR system. The reporter could affect its performance in two ways. First, the high or low modification level of the reporter site could result in biased performance in detecting positive or negative regulators. For example, in this study, our m⁵C site from FURIN genes has a very high modification rate. At this level, it would be much more sensitive in finding the decrease of methylation rate, therefore be much easier to discover the methyltransferase than potential demethylase if any in this case. In contrast, the use of reporter site with low modification rate would not be preferable in identifying methyltransferase. Second, except writer and eraser, most regulators may modulate the modification level through binding to the cis-regulatory elements, which are not necessarily in direct vicinity of the target site. A reporter constructs with a limited length might not be able to include all the relevant cis-elements. Consequently, we would fail to identify the regulators with binding sites missed in the reporter. On the other hand, the CIGAR-seq vector itself may contain artificial regulatory sequences affecting the modification of reporter site, which could result in the assay-specific artifacts. Therefore, subsequent careful validation with endogenous sites would be essential when working with the CIGAR-seq. The choice of CRISPR system could also have an effect. The screening based on CRISPR/Cas9 system, as applied in this study, would have limitations in finding potential regulators that are essential for cell survival and/or proliferation (e.g., METTL3/METTL14) since the gRNAs targeted at those essential genes would be largely depleted in the final sequencing library. This problem could be potentially alleviated by adopting CRISPRi or CRISPRa systems. In the future, with further improvements in CRISPR system and development of more sequencing-based readout with high precision, CIGAR-seq will become a versatile tool for systematic discoveries of players in multiple layer of RNA-based post-transcriptional gene regulation.

# Materials and Methods

## Reagents and Tools table

| Reagent/resource | Reference or source | Identifier or catalog number |
|---|---|---|
| Experimental models | | |
| HAP1 cells (*Homo sapiens*) | Horizon discovery | C631 |
| HEK-293T cells (*H. sapiens*) | ATCC | CRL-11268 |
| Recombinant DNA | | |
| LentiCas9-Blast | Addgene | #52962 |
| CROPseq-Guide-Puro | Addgene | #86708 |
| pLKO.1-puro | Addgene | #10878 |
| pLKO.1-blast plasmid | This study | N/A |
| psPAX2 | Addgene | #12260 |
| pMD2.G | Addgene | #12259 |
| Antibodies | | |
| Rabbit anti-NSUN2 | Proteintech | 20854-1-AP |
| Rabbit anti-NSUN6 | Proteintech | 17240-1-AP |
| Mouse anti-GAPDH | TransGen Biotech | HC301-01 |
| Goat anti-mouse IgG-HRP | Santa cruz biotechnology | sc-2005 |
| Goat anti-rabbit IgG-HRP | Santa cruz biotechnology | sc-2004 |
| Chemicals, enzymes, and other reagents | | |
| TRIzol® Reagent | Ambion | 15596026 |
| HiScript III 1$^{st}$ Stand cDNA Synthesis Kit | Vazyme Biotech | R312-02 |
| Hieff qPCR SYBR Green Master Mix | Yeasen | 11201ES08 |
| VAHTS mRNA Capture Beads | Vazyme Biotech | N401 |
| HiScript II Q Select RT SuperMix | Vazyme Biotech | R233-01 |
| HifairTM II 1st Strand cDNA Synthesis Kit | Yeasen | 11121ES60 |
| Agilent RNA 6000 Pico Kit | Agilent | NC1711873 |
| VAHTS Stranded mRNA-seq Library Prep Kit | Vazyme Biotech | NR602-01 |
| High Sensitivity DNA Kit | Agilent | 5067-4626 |
| Electrocompetent Stbl3 cell | Weidi Biotechnology | DE1046 |
| PEI | Polysciences | 23966-2 |
| BCA | Beyotime | P0011 |
| Polyvinylidene difluoride membranes | Immobilon-P | IPVH00010 |
| Pierce™ ECL Western Blotting Substrate | Thermo | 32209 |
| EZ RNA methylation kit | Zymo research | R5002 |
| RMPI1640 medium | Gibco | 22400089 |
| FBS | Gibco | 10270106 |
| P/S | Gibco | 15070063 |
| Oligonucleotides | | |
| Oligonucleotides | This study | Dataset EV3 |

## Methods and Protocols

### Cell culture and gene manipulation

HAP1 cell was obtained from Horizon discovery and cultured in RMPI1640 medium (Gibco) with 10% FBS (Gibco) and 1% P/S (Gibco) at 37°C with 5% $CO_2$. Cas9-expressing HAP1 cell line was established by using lentiCas9-Blast plasmid (Addgene). To generate NSUN2-KO, NSUN6-KO, and NSUN2/6-dKO clonal HAP1 cells, Cas9-expressing HAP1 cells were transduced with CROP-seq (Addgene) virus expressing following gRNAs: gNSUN2, #1; gNSUN6, #2. NSUN6

knockdown mediated by shRNA was performed using pLKO.1-blast plasmid (modified from pLKO.1-puro) with following shRNAs: shControl, #3; shNSUN6-1, #4; shNSUN6-2, #5.

### RT-qPCR

Total RNA was extracted by TRIzol® Reagent (Ambion). First-stand cDNA was synthesized using HiScript III 1st Stand cDNA Synthesis Kit (Vazyme). Quantitative PCR was performed by Hieff qPCR SYBR Green Master Mix (Yeasen) and the BIO-RAD real-time PCR system. Following primers were used to detect relative gene expression: NSUN6-F, #6; NSUN6-R, #7; GAPDH-F, #8; GAPDH-R, #9.

### CIGAR-seq vector with $m^5C/m^1A$ reporters and individual gRNA

Sequence flanking $m^5C$ site of FAM129B was amplified by forward primer #10 and reverse primer #11 from genomic DNA; $m^5C$ site of FURIN by forward primer #12 and reverse primer #13, and $m^1A$ site of MALAT1 by forward primer #14 and reverse primer #15. Amplified products were used to replace WPRE cassette in CROP-seq vector (Addgene) by ClonExpress II One Step Cloning Kit (Vazyme). Afterwards, following gRNAs were inserted at BsmBI sites to knockout individual genes: gControl, #16; gNSUN2, #1; gNSUN6, #2; gTRMT6, #17; gTRMT61A, #18.

### $m^5C$ detection by bisulfite conversion followed by sanger sequence

Total RNA was extracted by TRIzol® Reagent (Ambion). mRNA was enriched using VAHTS mRNA Capture Beads (Vazyme). 200 ng mRNA was converted by EZ RNA methylation kit (Zymo Research) according to the manufacturer's protocol with minor modification. More specifically, mRNA was incubated at 70°C for 10 min, and 60°C for 1 h. Converted RNA was then reverse transcribed into cDNA using HiScript II Q Select RT SuperMix (Vazyme).

To measure $m^5C$ rate in FURIN $m^5C$ reporter, target site was amplified using vector specific primer pair #19 and #20, and sanger-sequenced by primer #19. For $m^5C$ detection of endogenous $m^5C$ site in RPSA, target site was amplified using primer pair #21 and #22, and sanger-sequenced by primer #21.

### $m^1A$ detection based on mis-incorporation during reverse transcription

1 μg total RNA was reverse transcribed by HifairTM II 1st Strand cDNA Synthesis Kit (Yeasen). The region flanking $m^1A$ site was amplified by plasmid specific primer pair #23 and #24. The mismatch site was measured by sanger sequencing using primer #23.

### mRNA-BisSeq

The quality of 500 ng bisulfite-treated mRNA (see above) was assessed using Agilent RNA 6000 Pico Kit (Agilent) and then subjected to NGS libraries preparation using VAHTS Stranded mRNA-seq Library Prep Kit (Vazyme). The library quality was assessed using High Sensitivity DNA Kit (Agilent). Paired-end sequencing (2 × 150 bp) was performed with Illumina NovaSeq 6000 System by Haplox genomics center.

### Generation of CIGAR-seq vector pool with a FURIN $m^5C$ reporter

A gRNA library containing 4,975 gRNA targeting 829 RBP (Dataset EV1) was synthesized by GENEWIZ and cloned into CROP-seq vector (Addgene) at BsmBI sites. For measuring the complexity of the gRNA library, the region harboring gRNA sequence was amplified with primer pair #25 and #26 for NGS. Afterwards, the FURIN $m^5C$ reporter was amplified and used to replace WPRE cassette using ClonExpress II One Step Cloning Kit (Vazyme). During cloning of CIGAR-seq vector pool, electrocompetent Stbl3 cells (Weidi Biotechnology) were always used.

### CIGAR-seq viral package

HEK-293T cells were plated onto 15 cm plates at 40% confluence. The next day, cells were transfected with PEI (Polysciences) using 15 μg of CIGAR-seq vector, 15 μg of psPAX2 (Addgene), and 22.5 μg of pMD2.G (Addgene). Supernatant containing viral particles were harvested at 48 and 96 h and purified with 0.45 μm filter.

### Genetic screen for novel $m^5C$ regulators

1   First day, $2 × 10^8$ HAP1 cells were transduced with CIGAR-seq viral particles (MOI = 0.3).
2   After 24 h, HAP1 cells were treated with 1 μg/ml of Puromycin.
3   Puromycin resistant cells were cultured for additional seven days in medium containing 1 μg/ml of Puromycin.
4   Afterwards, $2 × 10^8$ HAP1 cells were collected for RNA extraction by TRIzol® reagent (Ambion).
5   All of extracted RNA was used to enrich poly(A)+ mRNA by VAHTS mRNA Capture Beads (Vazyme).
6   Enriched mRNA was quantified by Agilent RNA 6000 Pico Kit (Agilent).
7   A total of 100 ng mRNA was bisulfite converted by EZ RNA methylation kit (Zymo), then reversed transcribed into cDNA using HiScript II Q Select RT SuperMix (Vazyme).
8   All synthesized cDNA was used as template to PCR-amplify CIGAR-seq NGS library with primer pair #27 and #28.
9   Paired-end sequencing (2 × 150 bp) was performed with Illumina NovaSeq 6000 System by Haplox genomics center.

### Western blotting

HAP1 cells were collected and lysed by RIPA buffer (150 mM NaCl, 50 mM Tris, 1% EDTA, 1% NP40, 0.1% SDS). Lysate was incubated at 4°C for 30 min, then sonicated with 10 cycles (30 s On /30 s Off), and then centrifuged at 15,000 $g$ for 15 min at 4°C. The total protein concentration was measured by BCA (Beyotime). 60 μg total protein was loaded and separated on the 10% SDS–polyacrylamide gel. The protein on the gel was transfected to the polyvinylidene difluoride membranes (Immobilon-P). The membrane was incubated with primary antibody and horseradish peroxidase–conjugated secondary antibody, and then proteins were detected using the Pierce™ ECL Western Blotting Substrate (Thermo) by BIO-RAD ChemiDoc™ XRS+ system. The following antibodies were used for western blotting: NSUN2 (Proteintech), NSUN6 (Proteintech), GAPDH (TransGen Biotech), goat anti-mouse IgG-HRP (Santa cruz biotechnology), and goat anti-rabbit IgG-HRP (Santa cruz biotechnology).

### Computational methods

#### CIGRA-seq data analysis

CIGRA-seq NGS data consists of paired-end reads. Read1 contains the sequence of $m^5C$ reporter site while read2 consists of the gRNA sequence. Raw fastq data were first trimmed using fastp (Chen et al, 2018) to remove low-quality bases (-A -w 12 --length_required 30 -q

30). Then, the clean read pairs were parsed using a custom script based on pysam package. Specifically, gRNA sequence in read2 was extracted by regex module using regular expression ((CAACTT AACTCTTAAAC[ATCG]{20}CA){s<=1}). m⁵C reporter sequence was extracted in the similar way ((GTTATTT[TC]{1}TTTAAGG) {s<=1}). At most, 1 substitution was allowed during the pattern searching. Read pairs with both reads containing the matched pattern sequences and the m⁵C sites being C or T were kept for further analysis. Then for each gRNA sequence, the number of supported reads with reporter site being C (m⁵C) or T was calculated, and the number of C reads divided by the sum of C and T reads represented the m⁵C level. Only the extracted gRNA sequences that match exactly with the RBP gRNA sequences (Dataset EV1) were kept for further analysis.

To identify the high-confident candidate genes that regulate m⁵C level, information of multiple gRNAs of the same genes was combined using the Stouffer's method. gRNAs with no more than 20 supported reads were filtered out. Genes with only one gRNA detected were filtered out. Then, given a gene i, the m⁵C level of reporter site correspondence to gRNA j is $X_{i,j}$, m⁵C level was converted to $Z$-score and $P$ value $P_{i,j}$ was calculated under normal distribution assumption. Then, a combined $P$-value for each gene $P_i$ was obtain using the weighted version of Stouffer's method, with the logarithmic scale of read count as weight for each gRNA. Finally, $P$-values of multiple tests were adjusted with Benjamini & Hochberg's method.

### mRNA-BisSeq data analysis

mRNA-BisSeq data generated in this study were analyzed following the RNA-m⁵C pipeline (Huang *et al*, 2019) (https://github.com/SYSU-zhanglab/RNA-m5C). Reference genomes (GRCh38) and gene annotation GTF file was downloaded from Ensemble (http://www.ensembl.org/info/data/ftp/index.html). Briefly, raw paired-end reads were trimmed using cutadapt (Martin, 2011) (-a AGATCGGAAGAGCA CACGTC -A AGATCGGAAGAGCGTCGTGT -j 12 -e 0.25 -q 30 --trim-n) and then Trimmomatic (Bolger *et al*, 2014) (SLIDINGWINDOW:4:25 AVGQUAL:30 MINLEN:36). Clean read pairs were aligned to both C-to-T and G-to-A converted reference genomes by HISAT2 (Kim *et al*, 2019). Unmapped and multiple mapped reads were then aligned to C-to-T converted transcriptome by Bowtie2 (Langmead & Salzberg, 2012), and the transcript coordinates were liftovered to the genomic coordinates. Reads from HISAT2 and Bowtie2 mapping were merged and filtered using the same criteria as in RNA-m⁵C pipeline. Bam file was transformed into pileup file (--trim-head 6 --trim-tail 6). Putative m⁵C sites were called using script m5C_caller_multiple.py inside RNA-m⁵C pipeline (with parameters -P 8 -c 20 -C 2 -r 0.05 -p 0.05 --method binomial). Default parameters of RNA-m⁵C scripts were used unless otherwise specified.

### NSUN6-dependent m⁵C sites

First, to determine a set of high-confident m⁵C sites in HAP1 cells, five replicates of mRNA-BisSeq data generated from the WT HAP1 cells were used. The criteria to determine the high-confident m⁵C sites were as follows: (i) coverage of the site being at least 20 reads in all five replicates; (ii) number of reads containing the unmodified C being at least 2 in all five replicates; (iii) the WT methylation level (the minimum methylation level from the five replicates) being at least 0.05. Then, to determine the NSUN6-dependent m⁵C sites, m⁵C level of the sites was at least 0.05 in WT HAP1 cells and less than

0.02 or 10% of the WT m⁵C level in NSUN6-KO HAP1 cells. NSUN2-dependent sites were defined based on the same criteria.

### Features of the m⁵C sites

The upstream and downstream 10 bp sequences flanking the m⁵C sites were extracted from the genome. Motif analysis was performed Using ggseqlogo (Wagih, 2017) R package.

## Data availability

All next-generation sequencing data were submitted to Gene Expression Omnibus under the accession number GSE157368 (https://www.ncbi.nlm.nih.gov/geo/query/acc.cgi?acc=GSE157368).

**Expanded View** for this article is available online.

## Acknowledgements

This work was supported by the Shenzhen-Hong Kong Institute of Brain Science-Shenzhen Fundamental Research Institutions (Grant No. 2019SHIBS0002), Shenzhen Science and Technology Program (Grant No. KQTD20180411143432337, JCYJ20190809154407564, and JCYJ20180504165804015) and the National Natural Science Foundation of China (Grant No. 31701237, 31900431 and 31970601). The authors acknowledge the Center for Computational Science and Engineering of SUSTech for the support on computational resource and acknowledge the SUSTech Core Research Facilities and Guixin Ruan for technical support.

## Author contributions

WC and LF developed the concept of the project. WW, LF, LZ, DG, JY, and YT designed and performed experiments. GL performed bioinformatic analysis. WZ, MZ, DD, ZS, QZ, and ZD assisted in performing experiments. WC, LF, GL, YH, WW, JL, HC, and WS reviewed and discussed results. WC, LF, GL, and WW wrote the manuscript.

## Conflict of interest

The authors declare that they have no conflict of interest.

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
