## [Review Process File · Molecular Systems Biology]

CIGAR-seq, a CRISPR/Cas-based method for unbiased screening of novel mRNA modification regulators

Liang Fang, Wen Wang, Guipeng Li, Li Zhang, Jun Li, Diwen Gan, Jiao Yang, Yisen Tang, Zewen Ding, Min Zhang, Wenhao Zhang, Daqi Deng, Zhengyu Song, Qionghua Zhu, Huanhuan Cui, Yuhui Hu, and Wei Chen

DOI: [10.15252/msb.202010025](https://doi.org/10.15252/msb.202010025)

Corresponding author(s): *Wei Chen (chenw@sustc.edu.cn)* , *Liang Fang (fangl@sustech.edu.cn)*

Review Timeline:

Submission Date:	25th Sep 20
Editorial Decision:	26th Oct 20
Revision Received:	28th Oct 20
Editorial Decision:	3rd Nov 20
Revision Received:	4th Nov 20
Accepted:	4th Nov 20

Editor: Maria Polychronidou

Transaction Report:

Thank you again for submitting your work to Molecular Systems Biology. We have now heard back from the three referees who agreed to evaluate your study. Overall, the reviewers acknowledge that the proposed method seems interesting. However, they raise a series of concerns, which we would ask you to address in a major revision.

I think that the recommendations of the referees are clear and relatively straightforward to address, and I therefore see no need to repeat any of the points listed below. Please let me know in case you would like to discuss in further detail any of the issues raised. All issues raised by the referees would need to be satisfactorily addressed.

On a more editorial level, we would ask you to address the following.

REFeree REPORTS

Reviewer #1:

The authors developed a novel method named CIGAR-seq to screen mRNA modification enzymes. They selected m5C and m1A as examples. Using this novel CIGAR-seq approach, they identified NSUN6 as mRNA m5C methyltransferase., and NSUN6 and NSUN2 have different subsets of mRNA m5C sites. This work is interesting and provides useful method to study RNA modification. I am happy to recommend its publications after addressing the following minor issues.

- 1- Compared with Bisulfite method, CIGAR-seq might favor to mRNA targets with relative high level of m5C sites, thus I can see why the authors find the numbers of m5C sites are lower than that of previous published m5C sites. This should be included in the discussion.
- 2- Two previous results showed Nsun6 does not affect mRNA m5C level. I believe this could be caused by depletion approaches. Others used knock-down, but this work applied knock-out. The authors are encouraged to discuss this potential difference in the discussion section.
- 3- Yang X. Cell 2017 has been listed twice in the reference list.

Reviewer #2:

Fang et al. developed CIGAR-seq, a method that combines pooled CRISPR screen and reporters with RNA modification readout, and applied that to identify a novel m5C methyltransferase NSUN6. The manuscript is very focused and clearly written, providing strong evidence that NSUN6 is likely to be the only other m5C methyltransferase besides NSUN2 that was previously found. I noticed that there are two other groups that independently identified NSUN6 (published in BioRxiv), which would further validate the findings by this and other groups.

<https://www.biorxiv.org/content/10.1101/2020.10.01.320036v1>

<https://www.biorxiv.org/content/10.1101/2020.10.03.324707v1>

<https://www.biorxiv.org/content/10.1101/2020.10.03.324715v1> (this work)

Although I have some minor concerns listed below, I highly recommend acceptance of this work to be published in MSB.

1. While it is great to identify NSUN6 as the top hit, I am curious about a few other hits shown in Fig. 1D. I do not see the raw data from the screen. What are these hits - are they regulators of NSUN6?
2. The authors mentioned in their manuscript: "However, two previous studies did not show significant m5C changes in mRNA after NSUN6 perturbation in HeLa cells (Huang et al., 2019, Yang et al., 2017a)." This makes me wonder if this is due to technical challenges of bisulfite sequencing or biological differences in different cell types. I would like to see some experimental confirmation, and minimally some discussion of speculations.

Reviewer #3:

This is an interesting manuscript that describes an approach for using CRISPR to discover the enzyme that mediates the formation of a nucleotide modification within a given selected sequence. This approach is applied to a variety of sequences that are known to contain m5C in mRNA. For a site that is mediated by the enzyme NSUN2, the new CRISPR approach was able to successfully predict this enzyme. However for another site which contains m5C mediated by an unknown enzyme, this study may be important discovery that this was mediated by NSUN6. This is a new finding because this enzyme has not been previously linked as mediating m5C sites for epitranscriptome. Frankly, this entire manuscript could have been written from the perspective of discovering this new enzyme. Regardless, there are important basic science implications by discovering this enzyme in addition to the new method that has been described here.

There are limitations to this method. Most importantly, the modified nucleotide has to be able to incorporate or introduce a signature mutation. This is not always the case. Secondly, in most cases, it's pretty clear which enzyme makes the modified nucleotide. There are some exceptions, and the authors have clearly identified a good example.

Overall, this is nicely done, and the subsequent experiments to globally assess which m5C sites performed by which enzyme were nice. I think this is overall a good study. I have no major comments.

I have the following minor comments:

1. METTL14 is not a methyltransferase. The authors refer to METTL14 as a methyltransferase as proposed by He and colleagues (Liu et al 2013, doi: 10.1038/nchembio.1432). However it is now known that these METTL14 preparations prepared in insect cells were contaminated with insect METTL3 (see DOI: 10.1016/j.molcel.2016.05.041). Thus METTL14 is not an enzyme. The better reference would be Wang et al. 2014 (<https://doi.org/10.1038/ncb2902>). This paper was published at the same time as the earlier study and correctly proposed that METTL14 was part of a complex with METTL3, and the complex mediates m6A formation.
2. There are some spelling errors especially the word reverse is spelled incorrectly several times.

Point-by-point response to the referees' comments**General response to the reviewers:**

We thank the three reviewers for their time and appreciate their constructive comments. During the revision, we have carefully addressed all of them, as shown below marked in red.

Reviewer #1 (Comments to the Author):

The authors developed a novel method named CIGAR-seq to screen mRNA modification enzymes. They selected m5C and m1A as examples. Using this novel CIGAR-seq approach, they identified NSUN6 as mRNA m5C methyltransferase, and NSUN6 and NSUN2 have different subsets of mRNA m5C sites. This work is interesting and provides useful method to study RNA modification. I am happy to recommend its publications after addressing the following minor issues.

1-Compared with Bisulfite method, CIGAR-seq might favor to mRNA targets with relative high level of m5C sites, thus I can see why the authors find the numbers of m5C sites are lower than that of previous published m5C sites. This should be included in the discussion.

Answer:

Indeed, in the first part of our study, to ensure a high sensitivity in identifying the enzyme mediating m⁵C modification, we intentionally selected a m⁵C site of high modification rate to construct the epitranscriptomic reporter.

In the second part of our study, to characterize the NSUN6- and NSUN2-in/dependent m⁵C sites, we set stringent criteria to avoid potential false discovery of m⁵C sites due to incomplete bisulfite conversion. For this purpose, a total of five replicates of mRNA-BisSeq data were generated from the WT HAP1 cells. The high-confident m⁵C sites were determined based on: (1) coverage of the site being at least 20 reads in all five replicates; (2) number of reads containing the unmodified C being at least 2 in all five replicates; (3) the WT methylation level (the minimum methylation level from the five replicates) being at least 0.05 (described in Methods). Whereas such stringent criteria would assure the high specificity of our findings, we would likely also miss some of the true m⁵C sites, particularly those with low modification rate. We added a discussion about this in the revised manuscript (line 183-186).

2-Two previous results showed Nsun6 does not affect mRNA m5C level. I believe this could be caused by depletion approaches. Others used knock-down, but this work applied knock-out. The authors are encouraged to discuss this potential difference in the discussion section.

Answer:

Indeed, we think likewise. 'Huang et al., 2019' used siRNA to knockdown NSUN6 to ~40% of its original level (shown by RT-qPCR and Western blotting, Fig S5 b&c), which might not be sufficient to cause global m5C change. 'Yang et al., 2017a' also used siRNA to knockdown NSUN6 (verified by RT-PCR, Fig S3d), and induced slight, though not significant, global m5C reduction (Fig S3b). In comparison, we used CRISPR/Cas-mediated knockout to completely remove NSUN6, which is much more sensitive in

revealing the gene function. We have discussed the potential cause of this discrepancy in the revised manuscript (line 130).

We also noted the same group, who published 'Huang et al., 2019', now also identified NSUN6 as m⁵C methyltransferase (<https://www.biorxiv.org/content/10.1101/2020.10.03.324707v1>).

3-Yang X. Cell 2017 has been listed twice in the reference list.

Answer:

Thanks for pointing this out. We corrected it accordingly.

Reviewer #2 (Comments to the Author):

Fang et al. developed CIGAR-seq, a method that combines pooled CRISPR screen and reporters with RNA modification readout, and applied that to identify a novel m⁵C methyltransferase NSUN6. The manuscript is very focused and clearly written, providing strong evidence that NSUN6 is likely to be the only other m⁵C methyltransferase besides NSUN2 that was previously found. I noticed that there are two other groups that independently identified NSUN6 (published in BioRxiv), which would further validate the findings by this and other groups.

<https://www.biorxiv.org/content/10.1101/2020.10.01.320036v1>

<https://www.biorxiv.org/content/10.1101/2020.10.03.324707v1>

<https://www.biorxiv.org/content/10.1101/2020.10.03.324715v1> (this work)

Although I have some minor concerns listed below, I highly recommend acceptance of this work to be published in MSB.

1. While it is great to identify NSUN6 as the top hit, I am curious about a few other hits shown in Fig. 1D. I do not see the raw data from the screen. What are these hits - are they regulators of NSUN6?

Answer:

We have supplemented a list of candidate genes in Dataset EV2. Among all the RBPs we screened, only NSUN6 showed very significant P-value ($-\log_{10}(\text{P-value}) = 4.3$, two orders of magnitude lower than the second-ranked gene), and all six gRNAs targeting NSUN6 reduced m⁵C level effectively. Besides NSUN6, we selected ZCCHC11 (3rd-ranked) and EIF3J (5th-ranked) with marginally significant P-value ($-\log_{10}(\text{P-value}) = 1.8$ and 1.5, respectively) and multiple gRNAs (six and five, respectively) showing effect on inhibiting m⁵C modification for validation. However, knockout of either ZCCHC11 or EIF3J did not reduce the m⁵C modification at all. We described this result in the revised manuscript (line 143-146).

Please note that we did not choose 2nd- and 4th-ranked gene for validation, because both of them only had two out of six gRNAs showing effect on decreasing m⁵C level.

2. The authors mentioned in their manuscript: "However, two previous studies did not show significant m5C changes in mRNA after NSUN6 perturbation in HeLa cells (Huang et al., 2019, Yang et al., 2017a)." This makes me wonder if this is due to technical challenges of bisulfite sequencing or biological differences in different cell types. I would like to see some experimental confirmation, and minimally some discussion of speculations.

Answer:

We think the discrepancy between our study and other two is mainly due to the different gene silencing methods. Please see the address to Reviewer #1.

Reviewer #3 (Comments to the Author):

This is an interesting manuscript that describes an approach for using CRISPR to discover the enzyme that mediates the formation of a nucleotide modification within a given selected sequence. This approach is applied to a variety of sequences that are known to contain m5C in mRNA. For a site that is mediated by the enzyme NSUN2, the new CRISPR approach was able to successfully predict this enzyme. However, for another site which contains m5C mediated by an unknown enzyme, this study may be important discovery that this was mediated by NSUN6. This is a new finding because this enzyme has not been previously linked as mediating m5C sites for epitranscriptome. Frankly, this entire manuscript could have been written from the perspective of discovering this new enzyme. Regardless, there are important basic science implications by discovering this enzyme in addition to the new method that has been described here.

There are limitations to this method. Most importantly, the modified nucleotide has to be able to incorporate or introduce a signature mutation. This is not always the case. Secondly, in most cases, it's pretty clear which enzyme makes the modified nucleotide. There are some exceptions, and the authors have clearly identified a good example.

Overall, this is nicely done, and the subsequent experiments to globally assess which m5C sites performed by which enzyme were nice. I think this is overall a good study. I have no major comments.

I have the following minor comments:

1. METTL14 is not a methyltransferase. The authors refer to METTL14 as a methyltransferase as proposed by He and colleagues (Liu et al 2013, doi: 10.1038/nchembio.1432). However it is now known that these METTL14 preparations prepared in insect cells were contaminated with insect METTL3 (see DOI: 10.1016/j.molcel.2016.05.041). Thus METTL14 is not an enzyme. The better reference would be Wang et al. 2014 (<https://doi.org/10.1038/ncb2902>). This paper was published at the same time as the earlier study and correctly proposed that METTL14 was part of a complex with METTL3, and the complex mediates m6A formation.

Answer:

Thanks for pointing out our mistake. We have rewritten this part and cited the correct reference (Line 42-43).

2. There are some spelling errors especially the word reverse is spelled incorrectly several times.

Answer:

Thanks for pointing this out. We corrected them accordingly.

Thank you for sending us your revised manuscript. We think that the performed revisions have satisfactorily addressed the issues raised by the reviewers. I am therefore glad to inform you that we can soon formally accept the study for publication, pending some minor issues listed below:

2nd Authors' Response to Reviewers**4th Nov 2020**

The authors have made all the requested editorial changes.

Accepted**4th Nov 2020**

Thank you for performing these last few requested edits. We are now satisfied with the modifications made and I am pleased to inform you that your paper has been accepted for publication.

Corresponding Author Name: Wei Chen

Journal Submitted to: MSB

Manuscript Number: MSB-20-10025